

# Retrieval of Eddy Dissipation Rate from Derived Equivalent Vertical Gust included in Aircraft Meteorological Data Relay (AMDAR)

Soo-Hyun Kim[1], Hye-Yeong Chun[1], Jung-Hoon Kim[2], Robert D. Sharman[3], and Matt Strahan[4]

[1]Department of Atmospheric Sciences, Yonsei University, Seoul, South Korea
[2]School of Earth and Environmental Sciences, Seoul National University, Seoul, South Korea
[3]National Center for Atmospheric Research, Boulder, CO, USA
[4]NOAA/Aviation Weather Center, Kansas City, MO, USA

*Correspondence to*: Hye-Yeong Chun (chunhy@yonsei.ac.kr)

**Abstract.** Some of the Aircraft Meteorological Data Relay (AMDAR) data include a turbulence metric of the derived

equivalent vertical gust (DEVG), in addition to wind and temperature. As the cube root of the eddy dissipation rate (EDR) is

the International Civil Aviation Organization standard turbulence reporting metric, we attempt to retrieve the EDR from the

DEVG for more reliable and consistent observations of aviation turbulence globally. Using the DEVG in the AMDAR archived

from October 2015 to September 2018 covering a large portion of the Southern Hemisphere and North Pacific and North

Atlantic Oceans, we convert the DEVG to the EDR using two methods, after conducting quality control procedures to remove

suspicious turbulence reports in the DEVG. The first method is to remap the DEVG to the EDR using a lognormal mapping

scheme, while the second one is using the best-fit curve between the EDR and DEVG developed in the previous study. The

DEVG-derived EDRs obtained from the two methods are evaluated against in situ EDR data reported by United States-operated

carriers. For two specified regions of the trans-Pacific Ocean and Europe, where both the DEVG-derived EDRs and in situ

EDRs were available, the DEVG-derived EDRs obtained by the two methods are generally consistent with in situ EDRs, with

slightly better statistics by the first method than the second one. This result is encouraging for extending the aviation turbulence

data globally with the single preferred EDR metric, which will contribute to the improvement of global aviation turbulence

forecasting as well as to the construction of the climatology of upper-level turbulence.

## 1 Introduction

Turbulence observations are routinely provided verbally by pilots in the form of pilot reports (PIREPs). There may

be an uncertainty in the intensity, timing, and location of turbulence encounters in PIREPs (Schwartz, 1996; Sharman et al.,

2006, 2014), as the turbulence intensity in PIREPs is determined by a pilot's subjective measure of the aircraft response to

turbulence. Although PIREPs provide subjective categorized turbulence intensity scales (null, light, moderate, and severe), the

interpretation is aircraft dependent and null reports of turbulence events are not routine, therefore PIREPs are not adequate for

constructing reliable maps of turbulence levels. To address this deficiency, the automated objective aircraft-based reports of

turbulence are essential.





The Aircraft Meteorological Data Relay (AMDAR) system has been developed and operated by the World Meteorological Organization (WMO) as an operational observing system of automated aircraft weather observations. Given that the AMDAR data can provide routinely global atmospheric observations ranging from the surface to the upper air, these AMDAR data have been widely applied for monitoring and predicting weather systems and improving numerical weather prediction (NWP) models (Moninger et al., 2003). In addition to temperature and wind that are mandatory variables to report, two turbulence metrics are recommended to be included in the AMDAR data as measures of turbulence (WMO, 2003): the cube root of the eddy dissipation rate (EDR) (Sharman et al., 2014) and the derived equivalent vertical gust velocity (DEVG) (Hoblit, 1988).

The DEVG was introduced by Pratt and Walker (1954) and approximated to simplify the implementation (Sherman, 1985; Truscott, 2000) as:

$$DEVG \ (\text{m s}^{-1}) = \frac{Am|\Delta n|}{V_c}, \tag{1}$$

where parameter $A$ is the aircraft-specified parameter, $m$ is aircraft mass, $\Delta n$ is the maximum value of the deviation of vertical acceleration from 1 $g$ over a specified time interval, and $V_c$ is the calibrated air speed. For aircraft types, parameter $A$ can be approximated as

$$A = \overline{A} + c_4(\overline{A} - c_5)\left(\frac{m}{\overline{m}} - 1\right), \tag{2}$$

$$\overline{A} = c_1 + \left(\frac{c_2}{c_3 + H}\right), \tag{3}$$

where $H$ is the altitude in kft, $\overline{m}$ is the reference mass of the aircraft, and $c_1$, $c_2$, $c_3$, $c_4$, and $c_5$ are empirical constants dependent on the aircraft type that were given in Truscott (2000).

Due to the empirical parameters such as $c_1$, $c_2$, $c_3$, $c_4$, and $c_5$ in Eqs. (2) and (3), the DEVG could still include some uncertainties, which are assumed to be negligible after the rigorous quality control (QC) procedures in the current study. It is also noted that the DEVG does not consider the impact of pitch damping due to the autopilot (WMO, 2003; Kim et al., 2017). Since the DEVG can contain misleading values during the ascent and descent phases, previous studies have only considered the cruise-level DEVG values (e.g., Gill, 2014; Kim and Chun, 2016; Meneguz et al., 2016; Kim et al., 2017). The turbulence information defined by the DEVG has been utilized in statistical analyses on aviation turbulence (e.g., Kim and Chun, 2016; Kim et al., 2017) and in evaluations of the performances of NWP-based turbulence forecasts (e.g., Gill, 2014; Gill and Buchanan, 2014; Kim and Chun, 2016). Currently, the DEVG algorithm has been implemented on several international air-carriers such as the Qantas, South African, British Airways, and other European-based airline aircraft.

The EDR is estimated using aircraft vertical acceleration or estimated vertical wind velocity (MacCready, 1964; Cornman et al., 1995; Haverdings and Chan, 2010; Sharman et al., 2014; Cornman, 2016). The vertical winds-based EDR algorithm developed by the National Center for Atmospheric Research (NCAR) (Sharman et al., 2014; Cornman, 2016) is currently implemented on some fleets of the United Airlines, Delta Air Lines, and South west Airlines, while that developed





by Haverdings and Chan (2010) is tested on some aircraft of Hong Kong-based airline. Although Haverdings and Chan (2010) estimated the EDR in a similar way to Cornman (2016), they adopted the different angle-of-attack calibration and different time window and this may cause a difference between two EDRs. The EDR is more useful than the DEVG for turbulence metric detection and forecasting applications (Sharman et al., 2014), given that the DEVG is not a direct turbulence intensity

metric but a gust-load transfer factor. Indeed, the International Civil Aviation Organization (ICAO) assigned EDR as the preferred and standard metric for turbulence reporting (ICAO, 2001, 2010; Sharman et al., 2014). The EDR has been widely used in evaluations of the performances of global turbulence forecasting systems (e.g., Pearson and Sharman, 2017; Sharman and Pearson, 2017; Kim et al., 2018; Lee and Chun, 2018), as well as in many case studies on turbulence (e.g., Trier et al., 2012; Bramberger et al., 2018; Trier and Sharman, 2018).

Because two aforementioned turbulence metrics have been reported from different airliners, the EDR covers most areas in the Northern Hemisphere (NH), while the DEVG has been reported over a large portion of the Southern Hemisphere (SH). To complement the limited availability of global turbulence observations, in the current study, we attempt to convert the DEVG of the AMDAR data to the EDR to obtain more reliable and consistent observations for aviation turbulence. This will lead to improve the verification of global aviation turbulence forecasts as well as global climatology of aviation turbulence.

The relationship between the EDR and DEVG has been studied using flight data (e.g., Stickland, 1998; Kim et al., 2017). Stickland (1998) conducted a direct comparison between a vertical acceleration-based EDR and DEVG time series of Qantas Airways Boeing 747 data over a 3-month period (from October to December 1997) and showed that the two turbulence metrics are roughly correlated; however, this study considered a limited data period and only one aircraft type. Kim et al. (2017) compared the EDR from some aircraft of Hong Kong-based airline (Haverdings and Chan, 2010) and the DEVG from

the same aircraft using a relatively long period (39 months from February 2011 to April 2014) data. Kim et al. (2017) developed the best-fit curves between the EDR and DEVG for Airbus and Boeing aircraft data, separately. Although it was not directly used for the conversion of the DEVG to the EDR, Sharman and Pearson (2017) suggested a methodology to convert various turbulence diagnostics to the EDR by assuming that the turbulence diagnostics follow a lognormal distribution at upper levels. Here we propose to use this technique to convert the DEVG to the EDR.

For homogenized global aviation turbulence observations, in the current study, we convert the DEVG to the EDR using two conversion methods, one based on Sharman and Pearson (2017) and the other based on Kim et al. (2017), using historical DEVG records in the AMDAR National Oceanic and Atmospheric Administration (NOAA)-archives (hereafter, DEVG) dataset for 36 months (October 2015–September 2018). This paper is organized as follows. In section 2, the descriptions of the DEVG data, QC procedures applied on the DEVG data, and the QC'd DEVG statistics are provided. In

section 3, the conversion methods from the DEVG to the EDR and the DEVG-derived EDR statistics are examined. In section 4, a summary and discussion are provided.



## 2 Data and methodology

The AMDAR data archived at NOAA include both the EDR and DEVG from October 2015 to September 2018. Ideally, DEVG-based data and EDR-based data would be implemented and reported by the same aircraft so that direct comparisons could be made; however, this was not the case for the current AMDAR data. Furthermore, due to route structure differences, the spatiotemporal coincidence between the AMDAR EDR and DEVG data from different but nearby aircraft could not be constructed. Therefore, only a statistical comparison is examined, rather than the one-to-one comparison between the EDR and DEVG.

### 2.1 DEVG data

Figure 1 shows the horizontal distribution of the number of the raw DEVG data collected over 36 months (from October 2015 to September 2018) above 15 kft accumulated within a 1°×1° horizontal grid box. The data before the QC procedures to turbulence information are referred to as the raw DEVG in the current study. Fig. 1 shows that the raw DEVG covers a large portion of the SH, Africa, Europe, and the trans-Pacific and trans-Atlantic Oceans. Given that in situ EDR represents a large portion of the NH (Sharman and Pearson, 2017; Kim et al., 2018), this raw DEVG can complement the SH turbulence information. Reporting time window at the cruising level is generally between 7 and 21 minutes, and each DEVG is the maximum value over each time window (Gill, 2016). The raw DEVG data in some of the NH (e.g., the trans-Pacific Ocean and equatorial region) indicate relatively low reporting time window compared with those in the SH.

Figure 2 shows the horizontal locations of turbulence encounters expressed in raw DEVG values. When the DEVG is classified using the thresholds of 2, 4.5, and 9 m s$^{-1}$ for light (LGT), moderate (MOD), and severe (SEV) turbulence severity, respectively (Truscott, 2000; Gill, 2014; Kim and Chun, 2016), the numbers (percentage) of null (NIL), LGT, MOD, and SEV turbulence are 6,821,802 (95.5%), 187,985 (2.63%), 10,273 (0.14%), and 123,320 (1.73%), respectively. It seems to have some unrealistic SEV turbulence reports along the entire flight routes over the regions of Australia, New Zealand, and Europe, indicating the need for more careful QC procedures on those reports.

Figure 3 shows the probability density functions (PDFs) of the raw DEVG values at altitudes above 15 kft over the globe, the NH, and the SH for the same period (36 months). The primary peak falls within relatively small DEVG values (less than 8 m s$^{-1}$), and the secondary peak falls within relatively large DEVG values (greater than 8 m s$^{-1}$). This bimodal distribution, which is more prominent in the NH (blue curve) than in the SH (red curve), is highly suspicious considering that Kim et al. (2017) showed that the PDFs of the DEVG have a unimodal distribution following a lognormal distribution.

To examine the regional PDFs of the raw DEVG, we choose the following eight regions: region 1 covers some of Europe, Africa, and Asia (8°S–57°N, 5°E–65°E), region 2 covers East Asia (2°N–45°N, 60°E–160°E), region 3 covers the trans-Pacific Ocean and North America (28°S–50°N, 70°W–178°W), region 4 covers the trans-North Atlantic Ocean (5°N–65°N, 10°W–68°W), region 5 covers the trans-Indian Ocean (15°S–70°S, 30°E–108°E), region 6 covers Australia and New





Zealand (0°S–45°S, 110°E–180°E), region 7 covers the trans-South Pacific Ocean (30°S–75°S, 70°E–178°E), and region 8 covers the trans-South Atlantic Ocean (42°S–4°N, 60°W–28°E).

Figure 4 shows the PDFs of the raw DEVG over these eight regions. As shown in Fig. 3, the PDFs of the DEVG in regions 1 and 6, covering Europe and Australia-New Zealand, respectively, show clear bimodal distributions. In contrast, the PDFs of the DEVG in regions 2–5 and 7–8 show unimodal distributions. The DEVG in regions 4, 7, and 8 does not include strong turbulence events (e.g., DEVG > 9 m s$^{-1}$). Commonly, Figs. 2–4 show that the raw DEVG may contain erroneous turbulence reports, which requires QC procedures to remove those erroneous turbulence reports from the raw DEVG.

**2.2 QC procedures**

In the QC procedures, the DEVG, longitude, latitude, altitude, and flight tail number are used. Notably, aircraft-related information, such as aircraft type and tail number, is limited in the AMDAR dataset, and time series of basic variables required for a DEVG calculation are not available. That is, because the raw DEVG data with the same tail number sometimes include multiple flights, the flight tail number is only used to separate individual flights.

Figure 5 shows a flow chart of the QC procedures. First, the raw DEVG data are redistributed into an individual file which has the same flight tail number. Second, data are considered to be erroneous according to the following criteria:

1) If the number of observations in the individual file is less than eight.

2) If, for the individual file, more than two SEV and more than six MOD turbulence events are counted within the spatiotemporal window, which is defined as a circle with 100 km radius, a time window of ±1 hour, and an altitude window of ±3 kft.

3) If there is only one reported SEV turbulence event, but no MOD turbulence event within a 200 km radius-circle and time window of ±1 hour.

4) If there is only one reported MOD turbulence event, but no LGT turbulence event within a 200 km radius-circle and time window of ±1 hour.

Applying the aforementioned QC procedures, only the QC'd DEVG data (hereafter, QCDEVG) are examined in the present study. The current QC procedures are designed to increase confidence in the observed turbulence events considering surrounding turbulence events. The ratio of SEV to MOD turbulence events is larger in the current study (> 2/6) than in other observational studies. For example, over South Korea, Kim and Chun (2011) showed 2.94% MOD and 0.08% SEV turbulence events from the PIREPs, while Kim and Chun (2016) showed 0.25% (0.33%) MOD and 0.04% (0.04%) SEV turbulence events from 1-minute aircraft data over the globe (East Asia) and 5.1% MOD and 0.34% SEV turbulence events from the PIREPs over East Asia. Nevertheless, in the current study, the spatial and temporal windows are empirically determined to satisfactorily remove suspicious turbulence reports from the raw DEVG data.

Figure 6 shows the horizontal locations of turbulence encounters according to the QCDEVG above 15 kft for 36 months (from October 2015 to September 2018). The QC procedures indicate that 6,269,077 (97.28%) NIL, 170,199 (2.64%) LGT, 5,380 (0.083%) MOD, and 32 (0.0005%) SEV turbulence events defined by the DEVG values are valid. Most of the



SEV turbulence events over Europe, Australia and New Zealand are discarded by the QC procedures. Many discarded turbulence observations over Australia and New Zealand are due to continuous SEV turbulence reports or single SEV turbulence reports without consecutive NIL, LGT, and MOD turbulence events (not shown), while those over Europe are due to a single SEV or MOD turbulence report of the 8 reports within an individual file. Relatively many SEV turbulence events over the trans-Pacific Ocean and trans-Indian Ocean pass the QC procedures and they are considered as valid turbulence reports. Some of the MOD and SEV turbulence reports coincide with the regions indicating high turbulence potential determined by the Ellrod1 index (Ellrod and Knapp, 1992) which is a conventional clear-air turbulence diagnostic (not shown).

## 2.3 Spatial statistics of the QCDEVG

Figure 7 shows the PDFs of the QCDEVG at altitudes above 15 kft over the globe, the NH, and the SH. As shown in Fig. 6, the secondary peaks in Fig. 3 are no longer apparent in Fig. 7. The SEV turbulence events defined by the DEVG values account for highly reduced percentages of $10^{-4}$ %. The PDFs of the QCDEVG indicate a unimodal distribution, which is consistent with Fig. 4 of Kim et al. (2017). The PDF for the NH indicates a relatively steep slope for low DEVG values compared with the PDF for the SH. Accordingly, the lognormal fitting, which will be shown in section 3, is conducted for the NH and SH, separately, as characteristics of the QCDEVG are hemisphere dependent.

Figure 8 shows the regional PDFs of the QCDEVG values over the eight regions shown in Fig. 4. The horizontal distribution of the number of the QCDEVG data accumulated within the 1°×1° horizontal grid box is also indicated in Fig. 8. Most DEVG reports are in the SH and along the narrow flight tracks over the trans-Atlantic Ocean and between Africa and Europe or Asia. The PDFs of the QCDEVG archived in regions 1 and 6 show unimodal distributions. The PDFs of the QCDEVG show quite similar distributions to those calculated using the raw DEVG for the six regions (regions 2–5 and 7–8). The QCDEVG data in regions 5–8 generally are concentrated in the low DEVG value compared with those in regions 1-4. Our focus is to remove suspicious turbulence reports within the limited aircraft-related information and to obtain a reasonable PDF indicating a unimodal distribution. In this regard, the quality of the QCDEVG is considered adequate for the EDR conversion.

## 3 Conversion of the QCDEVG to the EDR

The QCDEVG is now converted to the EDR using two methods (hereafter, DEVG-derived EDR), as EDR is the preferred turbulence forecast metric. The methods considered in the current study are based on Sharman and Pearson (2017) and Kim et al. (2017). Brief descriptions of the two methods are provided below.

## 3.1 EDR conversion using the lognormal mapping scheme

Considering that the distribution of observed EDR in the free atmosphere approximately follows a lognormal distribution (Nastrom and Gage, 1985; Frehlich, 1992; Cho et al., 2003; Frehlich and Sharman, 2004; Sharman et al., 2014; Kim et al., 2017), Sharman and Pearson (2017) proposed a statistical mapping equation applying NWP-based turbulence





diagnostics to the EDR. Assuming the lognormal property of turbulence diagnostics, the simplest mapping between a raw turbulence diagnostic $D$ and the EDR is provided by:

$$\ln(D^*) = \ln(\text{EDR}) = a + b\ln(D),\tag{4}$$

where $D^*$ is the remapped EDR value corresponding to the raw turbulence diagnostic $D$, slope $b$ is the ratio between the standard deviation (SD) of $\ln(\text{EDR})$ and SD of $\ln(D)$ [$b = \text{SDln(EDR)}/\text{SDln}(D) = C_2/\text{SDln}(D)$] and the intercept $a$ is the difference between the mean of $\ln(\text{EDR})$ and mean of $\ln(D)$ [$a = \langle\ln(\text{EDR})\rangle - b\langle\ln(D)\rangle = C_1 - b\langle\ln(D)\rangle$, where the angle brackets indicate the ensemble mean]. Here, $C_1$ and $C_2$ are the climatological values of the mean and SD of $\ln(\text{EDR})$, respectively, which are obtained from the lognormal fits to the EDR estimates of in situ equipped aircraft from 2009 to 2014.

To utilize this statistical mapping equation to obtain the DEVG-derived EDRs, the turbulence diagnostic $D$ is replaced with the DEVG value. Thus, Eq. (4) can be written as:

$$\ln(\text{DEVG}^*) = \ln(\text{EDR}) = a + b\ln(\text{DEVG}),\tag{5}$$

where $\text{DEVG}^*$ is the remapped EDR value corresponding to the QCDEVG value. The intercept $a$ and the slope $b$ can be written as:

$$a = \langle\ln(\text{EDR})\rangle - b\langle\ln(\text{DEVG})\rangle = C_1 - b\langle\ln(\text{DEVG})\rangle \text{ and}$$

$$b = \text{SDln(EDR)}/\text{SDln(DEVG)} = C_2/\text{SDln(DEVG)}\tag{6}$$

The parameters $C_1$ and $C_2$ for four different altitude bands (-2.248 and 0.4235 for altitudes of 0–10 kft, -2.578 and 0.557 for altitudes of 10–20 kft, -2.953 and 0.602 for altitudes of 20–45 kft, and -2.572 and 0.5067 for altitudes above 0 ft, respectively) are given in Sharman and Pearson (2017). Although the values $C_1$ and $C_2$ can be used for three different altitude ranges (one is the altitudes above 0 ft, another is the altitudes of 20–45 kft, and the other is the altitudes of 10–20 kft and 20–45 kft), the values for the upper levels of 20–45 kft are utilized in the current study, considering that the values $C_1$ and $C_2$ are not significantly altitude dependent. To obtain the mean and SD of $\ln(\text{DEVG})$, the values of the QCDEVG over the NH and SH are calculated for the lognormal fitting via the optimization function "fminsearch" in the MATLAB package (Lagarias et al., 1998; see also https://www.mathworks.com/help/matlab/ref/fminsearch.html). The EDR converted from this method is called for EDR-SP17, hereafter.

Figure 9 shows the lognormal fits (curves) applied to the PDFs (filled circles) of the QCDEVG values over the NH and SH (blue and red lines in Fig. 8, respectively). To obtain an optimized lognormal curve, some of the highest and lowest bins (open circles) of the QCDEVG are not used for the lognormal fits. The mean values of $\ln(\text{DEVG})$ over the NH and SH are -0.69926 and -1.4397 m s$^{-1}$, respectively, and the SDs of $\ln(\text{DEVG})$ over the NH and SH are 0.6956 and 0.7773 m s$^{-1}$, respectively. The mean and SD of $\ln(\text{DEVG})$ over the SH and NH are used for the EDR conversion (EDR-SP17).



### 3.2 EDR conversion using the prescribed best-fit function

Kim et al. (2017) investigated two turbulence indicators (the EDR and the DEVG) calculated by the algorithms using the time series of several variables recorded by Hong Kong-based airline flight data recorders for 39 months from February 2011 to April 2014. On a one-to-one basis, relationships between the EDR and DEVG are calculated for three different Boeing

(B) aircraft models (B747-400, B777-200, and B777-300) and three different Airbus (A) aircraft models (A320-200, A321-200, and A330-300), which developed the best-fit quadratic functions for Boeing and Airbus aircraft, separately.

The quadratic equations for the Boeing and Airbus aircraft data are as follows:

$$\text{DEVG}^* = \text{EDR} = 0.0031\left(\text{DEVG}^2\right) + 0.0286(\text{DEVG}) + 0.0114, \text{ for Boeing} \tag{7}$$

$$\text{DEVG}^* = \text{EDR} = 0.003\left(\text{DEVG}^2\right) + 0.0324(\text{DEVG}) + 0.0516, \text{ for Airbus} \tag{8}$$

where $\text{DEVG}^*$ is the converted EDR corresponding to the QCDEVG. Although two different DEVG-derived EDRs can be derived using the above two quadratic equations, the DEVG-derived EDR obtained from the quadratic equation for the Boeing aircraft (Eq. 7), which shows a high correlation between the EDR and DEVG, is considered exclusively in the current study. The EDR converted from this method is called for EDR-KCC17, hereafter.

### 3.3 Spatial statistics of the DEVG-derived EDRs

Table 1 shows the mean and SD of the natural logarithms of EDR-SP17 and EDR-KCC17, ln(EDR-SP17) and ln(EDR-KCC17), respectively, for the eight regions indicated by rectangles in Fig. 8. The mean and SD of the resultant DEVG-derived EDRs differ slightly among the eight specified regions. Nevertheless, regarding the mean of the natural logarithm of the EDR, EDR-SP17 (from -2.9986 to -1.8083 $m^{2/3}$ $s^{-1}$) is larger than EDR-KCC17 (from -3.9340 to -3.0691 $m^{2/3}$ $s^{-1}$) for all eight regions, with differences in magnitude ranging from 0.4788 to 1.2608 $m^{2/3}$ $s^{-1}$. For the SD of the natural logarithm of the

EDR, EDR-SP17 (from 0.3057 to 1.0538 $m^{2/3}$ $s^{-1}$) is larger than EDR-KCC17 (from 0.2196 to 0.6941 $m^{2/3}$ $s^{-1}$) for all eight regions, with differences in magnitude ranging from 0.0861 to 0.3597 $m^{2/3}$ $s^{-1}$.

Given that EDR-SP17 and EDR-KCC17 have different characteristics, validation of the two different methods is required. Accordingly, the EDRs estimated from in situ equipped aircraft implemented in some United States (US) commercial aircraft (Sharman et al., 2014; Cornman, 2016) are used as the reference data (hereafter, USEDR). The comparison between

25 the USEDR and DEVG-derived EDRs proposed in the current study for the same period (from October 2015 to September 2018) is conducted by comparing the mean and SD values of the natural logarithms of three different EDRs (EDR-SP17, EDR-KCC17, and USEDR) for the specified regions.

Figure 10 shows the horizontal distribution of the USEDR counts (reference data) at altitudes above 15 kft accumulated within a 1°×1° horizontal grid box from the same period (36 months) with the DEVG data. Compared with Fig.

8, the USEDR data mainly cover large portions of the NH, which include the flight routes of the trans-Pacific Ocean, North and South America, the trans-Atlantic Ocean, and Europe. To evaluate the feasibility of deriving the EDRs from the DEVG





using the two methods (EDR-SP17 and EDR-KCC17), the mean and SD of three different EDRs are calculated over the two specified regions represented by the rectangles in Fig. 10; one region covers some of Europe and the other covers the trans-Pacific Ocean, which includes flight routes between North America and Australia. Although there are much USEDR data (Fig. 10) over North America and the trans-Atlantic Ocean, unfortunately, the DEVG data (Fig. 8) over these two regions are insufficient for further analysis.

Figure 11 shows the PDFs of EDR-SP17, EDR-KCC17, and USEDR data over the two rectangles in Fig. 10 from October 2015 to September 2018. Over both Europe and the trans-Pacific Ocean, the distributions of the PDF of EDR-SP17 and USEDR are similar. Especially for the trans-Pacific Ocean region, the PDFs of EDR-SP17 and USEDR at values larger than ~0.22 $m^{2/3}$ $s^{-1}$ are in very good agreement. Over Europe (Fig. 11a), the values of EDR-SP17 are generally larger than those of EDR-KCC17 and USEDR, while over the trans-Pacific Ocean (Fig. 11b), EDR-SP17 and USEDR are similar. The EDR-KCC17 has a larger percentage of low EDR values ($< $ ~0.1 $m^{2/3}$ $s^{-1}$) compared to EDR-SP17 and USEDR in the two regions. For each PDF shown in Fig. 11, the root mean square error (RMSE) of the occurrence frequency of two different EDRs (EDR-SP17 and EDR-KCC17) is calculated with respect to that of the USEDR. Over Europe, the RMSE of EDR-SP17 is 0.0157, and that of EDR-KCC17 is 0.0441. Over the trans-Pacific Ocean, the RMSE of EDR-SP17 is 0.0504, and that of EDR-KCC17 is 0.0903, implying that the occurrence frequency of EDR-SP17 is relatively close to that of the USEDR. The PDFs of EDR-SP17 and USEDR generally follow lognormal distributions, whereas the PDF of EDR-KCC17 departs somewhat from a lognormal distribution especially at low EDR values ($< $ ~0.14 $m^{2/3}$ $s^{-1}$) (not shown). It is noted that the slight difference between the EDR calculations of Cornman (2016) and Haverdings and Chan (2010) might result in the observed difference in the EDR statistics and affect the DEVG-derived EDRs.

Table 2 shows the mean and SD of the natural logarithm of three different EDRs (EDR-SP17, EDR-KCC17, and USEDR) over Europe and the trans-Pacific Ocean. For the region of Europe, the mean values of ln(EDR-SP17) and ln(EDR-KCC17) are -2.2394 and -2.5674 $m^{2/3}$ $s^{-1}$, respectively, and the SDs of ln(EDR-SP17) and ln(EDR-KCC17) are 0.4782 and 0.3522 $m^{2/3}$ $s^{-1}$, respectively. For the trans-Pacific Ocean region, the mean values of ln(EDR-SP17) and ln(EDR-KCC17) are -2.0299 and -2.7384 $m^{2/3}$ $s^{-1}$, respectively, and the SDs of ln(EDR-SP17) and ln(EDR-KCC17) are 0.4136 and 0.2678 $m^{2/3}$ $s^{-1}$, respectively. The EDR-SP17 and USEDR generally have relatively close mean and SD values, which implies that the EDR-SP17 technique is more accurate at least in the current case. In our current limited study, the statistical properties between EDR-SP17 and USEDR appear slightly different, with higher intensities overall over Europe than the trans-Pacific Ocean. However, because the results are considered only over two regions, further evaluation of the two different methods for deriving EDRs from DEVG is required over different regions and longer period datasets.

## 4 Summary and discussion

In the current study, we convert the AMDAR provided turbulence indicator, the DEVG, to the EDR to obtain quantitative and consistent turbulence observations globally. We use the DEVG data archived in the NOAA AMDAR (raw



DEVG) data for 36 months (October 2015 to September 2018). In the raw DEVG data, there are many suspicious strong-intensity turbulence reports that cause bimodal distributions in the PDFs of the DEVG. To remove erroneous turbulence reports in the raw DEVG data, QC procedures are developed by applying optimally determined thresholds to the raw DEVG dataset. The QC'd DEVG values are converted to the EDR, which is the ICAO standard turbulence intensity metric. The conversion

of the DEVG to the EDR is conducted using two methods. Sharman and Pearson (2017) proposed a linear mapping equation assuming the lognormal property of raw turbulence diagnostics, while Kim et al. (2017) proposed the best-fit curve (quadratic equation) between the EDR and DEVG, based on the one-to-one comparison between the EDR and DEVG calculated using the same flight data. The PDFs of the resultant DEVG-derived EDRs from the two methods, referred to as EDR-SP17 and EDR-KCC17, are compared with those of the USEDR for the two regions covering Europe and the trans-Pacific Ocean. It is

found that EDR-SP17 has a relatively similar distribution with the USEDR at least for the current case.

The robust conversion of the DEVG to the EDR would improve the verification of turbulence forecasts globally and the investigation of global characteristics of aviation turbulence, as the USEDR data are still of limited availability globally (Fig. 10). Indeed, the characteristics of aviation turbulence over the NH have been investigated in many previous studies, while those over the SH have not, in part due to a lack of observational data. In this regard, qualified DEVG-derived EDRs can be

an important additional source of information globally, especially in most of the SH. Additionally, the DEVG data used in the current study can represent valuable observations for the evaluation of turbulence diagnostics related to convection (Kim et al., 2019), given that the DEVG data contain substantial turbulent information over the tropical region. Together with the existing the USEDR data over the NH, the DEVG-derived EDRs in the SH and tropical regions can be merged into a homogenized global turbulence information, which will contribute to improvement of global aviation turbulence forecasting

as well as to construction of global climatology of upper-level turbulence.

*Data availability.* The AMDAR data archived at NOAA are available at https://madis-data.ncep.noaa/gov/madisPublic1/data/archive.

*Author contributions.* SHK, HYC, and JHK designed the study. SHK prepared the original draft of the paper with contributions from HYC, JHK, RDS, and MS. Together, SHK, HYC, JHK, and RDS interpreted the results and reviewed and edited the paper.

*Competing interests.* The authors declare they have no conflict of interest.

*Acknowledgements.* This work was funded by the Korea Meteorological Administration Research and Development Program under Grant KMI 2018-07810. Additional acknowledgment is given to the University Corporation for Atmospheric Research (UCAR) and the National Centers for Environmental Prediction (NCEP) for allowing the first author to partake in the UCAR/NCEP Visiting Scientist Program.



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



**Table 1.** Values of the mean and standard deviation (SD) of the natural logarithms of EDR-SP17 and EDR-KCC17 over the eight selected regions indicated in Fig. 8, from October 2015 to September 2018. The unit is $m^{2/3} s^{-1}$. Note that EDR-SP17 and EDR-KCC17 are the DEVG-derived EDRs obtained using the methods of Sharman and Pearson (2017) and Kim et al. (2017), respectively.

(a) Mean

|  | region 1 | region 2 | region3 | region 4 | region 5 | region 6 | region 7 | region 8 |
|---|---|---|---|---|---|---|---|---|
| EDR-SP17 | -2.8144 | -2.6072 | -2.5612 | -2.9192 | -2.0771 | -2.9986 | -2.4930 | -1.8083 |
| EDR-KCC17 | -3.3845 | -3.2647 | -3.4031 | -3.3980 | -3.3083 | -3.9340 | -3.6562 | -3.0691 |

(b) Standard deviation

|  | region 1 | region 2 | region 3 | region 4 | region 5 | region 6 | region 7 | region 8 |
|---|---|---|---|---|---|---|---|---|
| EDR-SP17 | 0.8755 | 0.7308 | 0.6539 | 1.0538 | 0.5353 | 0.6090 | 0.5150 | 0.3057 |
| EDR-KCC17 | 0.6360 | 0.5240 | 0.5144 | 0.6941 | 0.4148 | 0.3955 | 0.4026 | 0.2196 |



Atmospheric
Measurement
Techniques

Discussions

**Table 2.** Values of the mean and SD of the natural logarithms of EDR-SP17, EDR-KCC17, and USEDR over (a) Europe and (b) the trans-Pacific Ocean routes indicated in Fig. 10, from October 2015 to September 2018. The unit is $m^{2/3} s^{-1}$.

(a) Europe

|  | EDR-SP17 | EDR-KCC17 | USEDR |
|---|---|---|---|
| Mean | -2.2394 | -2.5674 | -2.3258 |
| SD | 0.4782 | 0.3522 | 0.4118 |

(b) Trans-Pacific Ocean

|  | EDR-SP17 | EDR-KCC17 | USEDR |
|---|---|---|---|
| Mean | -2.0299 | -2.7384 | -2.3171 |
| SD | 0.4136 | 0.2678 | 0.4010 |



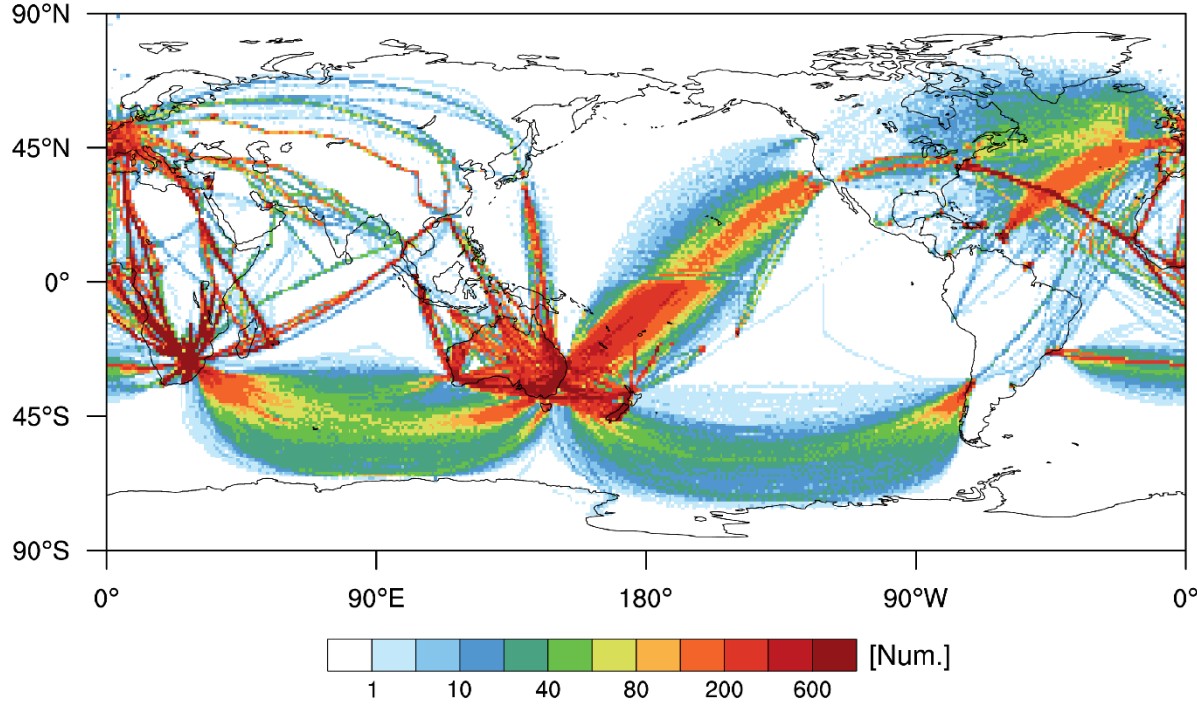

**Figure 1.** Horizontal distribution of the number of the raw DEVG data at altitudes above 15 kft, accumulated within a 1°×1° horizontal grid box from October 2015 to September 2018.





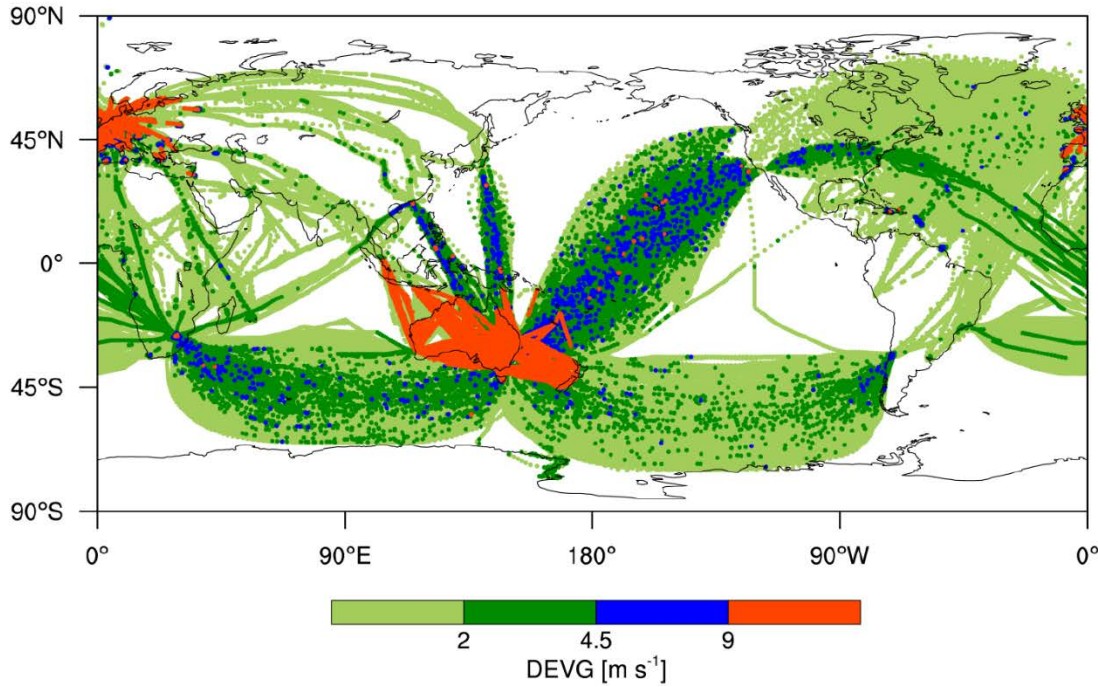

**Figure 2.** Horizontal locations of turbulence encounters expressed in raw DEVG values at altitudes above 15 kft from October 2015 to September 2018.

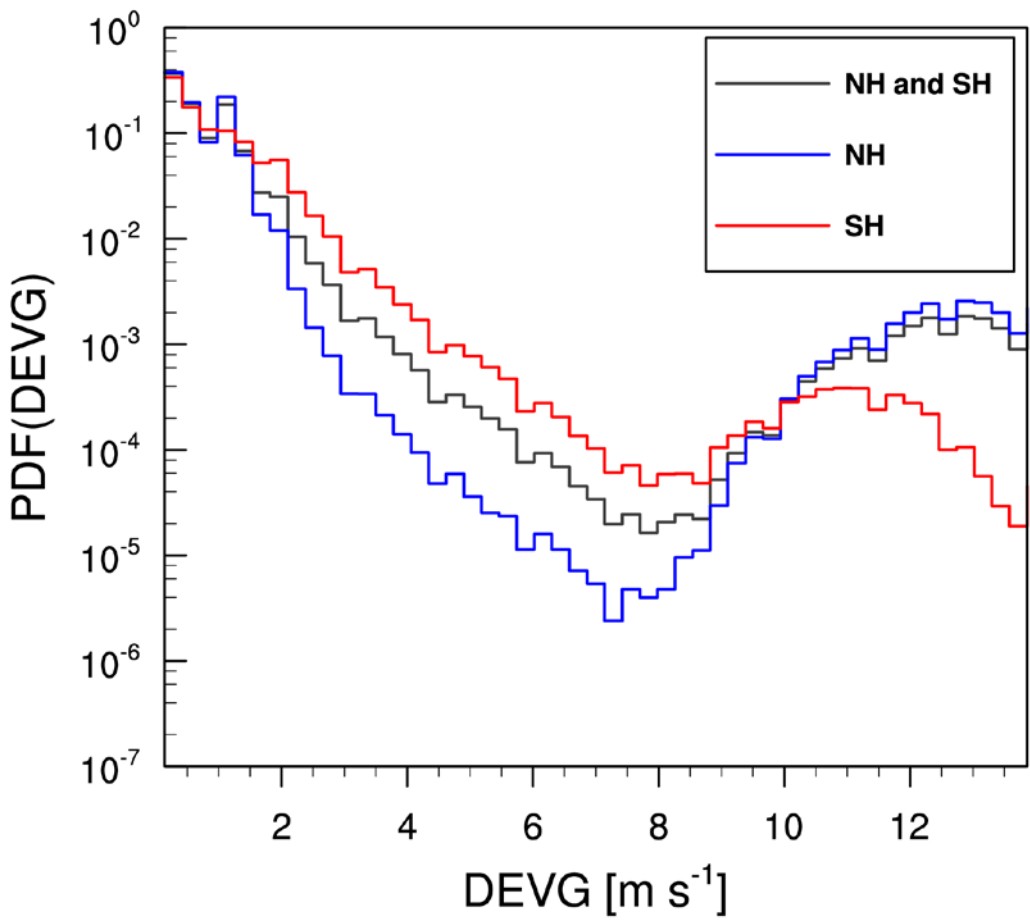

**Figure 3.** The probability density functions (PDFs) of the raw DEVG over the Northern Hemisphere (NH: blue line), the Southern Hemisphere (SH: red line), and the globe (NH and SH: black line) at altitudes above 15 kft from October 2015 to September 2018.







**Figure 4.** The PDFs of the raw DEVG over eight selected regions, which are indicated as rectangles in the global map, at altitudes above 15 kft from October 2015 to September 2018. The eight regions are located in 8S°–57°N, 5°E–65°E (region 1); 2°N–45°N, 60°E–160°E (region 2); 28°S–50°N, 70°W–178°W (region 3); 5°N–65°N, 10°W–68°W (region 4); 15°S–70°S, 30°E–108°E (region 5); 0°S–45°S, 110°E–180°E (region 6); 30°S–75°S, 70°E–178°E (region 7); and 42°S–4°N, 60°W–28°E (region 8).





**Figure 5.** The flow chart of quality control procedures.




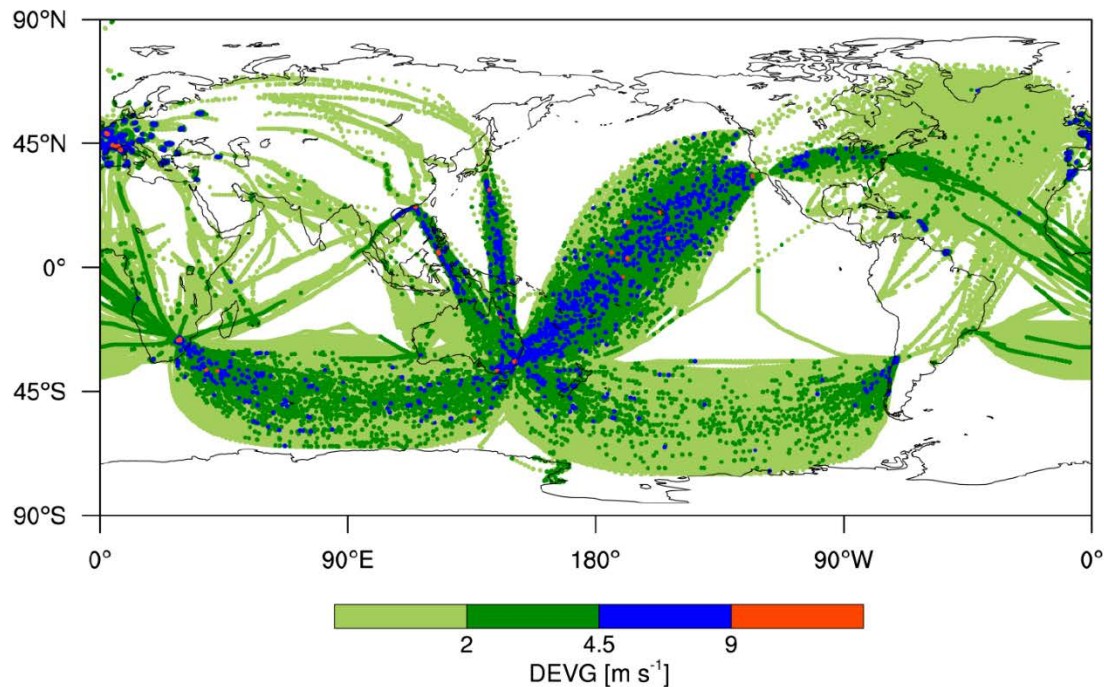

**Figure 6.** As in Fig. 2, except for the QCDEVG.



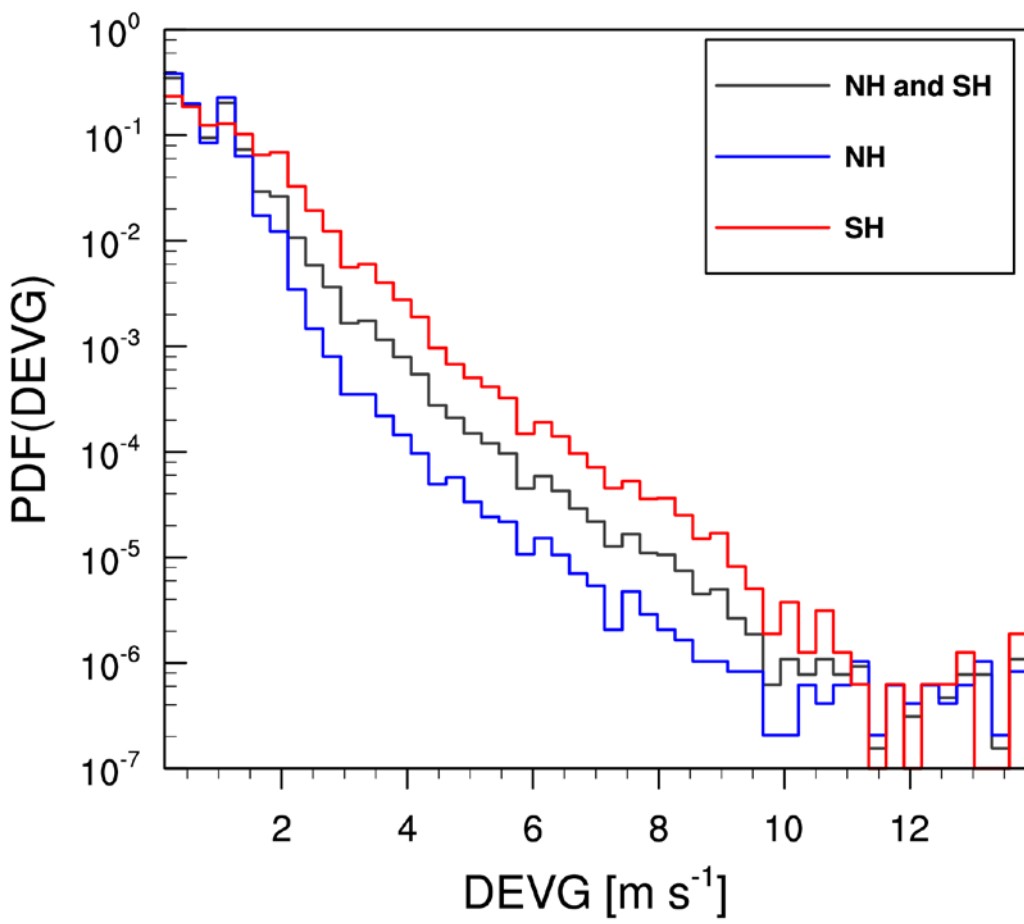

**Figure 7.** The PDFs of the QCDEVG over the globe (NH and SH: black line), NH (blue line), and SH (red line) at altitudes above 15 kft from October 2015 to September 2018.





**Figure 8.** Horizontal distribution of the number of the QCDEVG data at altitudes above 15 kft, accumulated within a 1°×1° horizontal grid box from October 2015 to September 2018. The PDFs of the QCDEVG over eight selected regions, superimposed on the global map by rectangles.



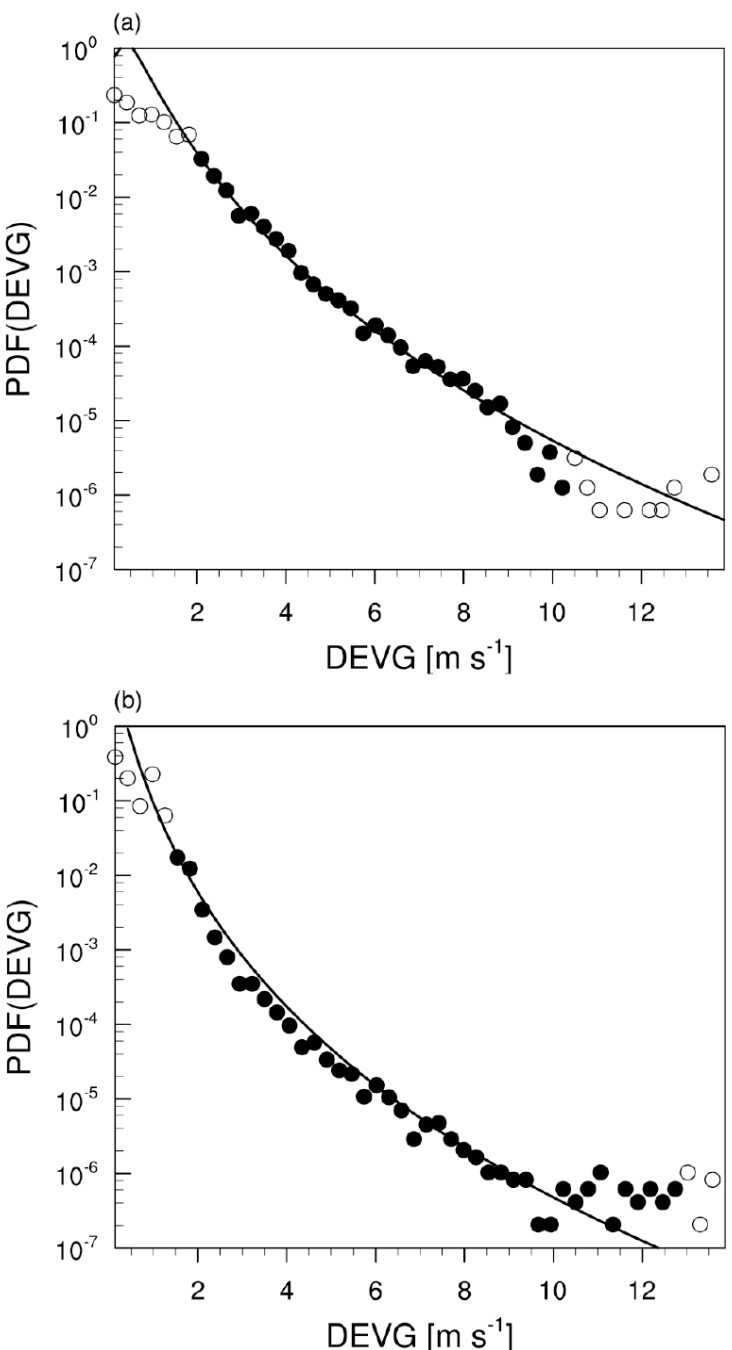

**Figure 9.** The PDFs (circles) of the QCDEVG and lognormal fit (continuous line) over the QCDEVG over the (a) NH and (b) SH. The filled circles indicate data that were used in the fit, and the open circles indicate data that are excluded from the fit.



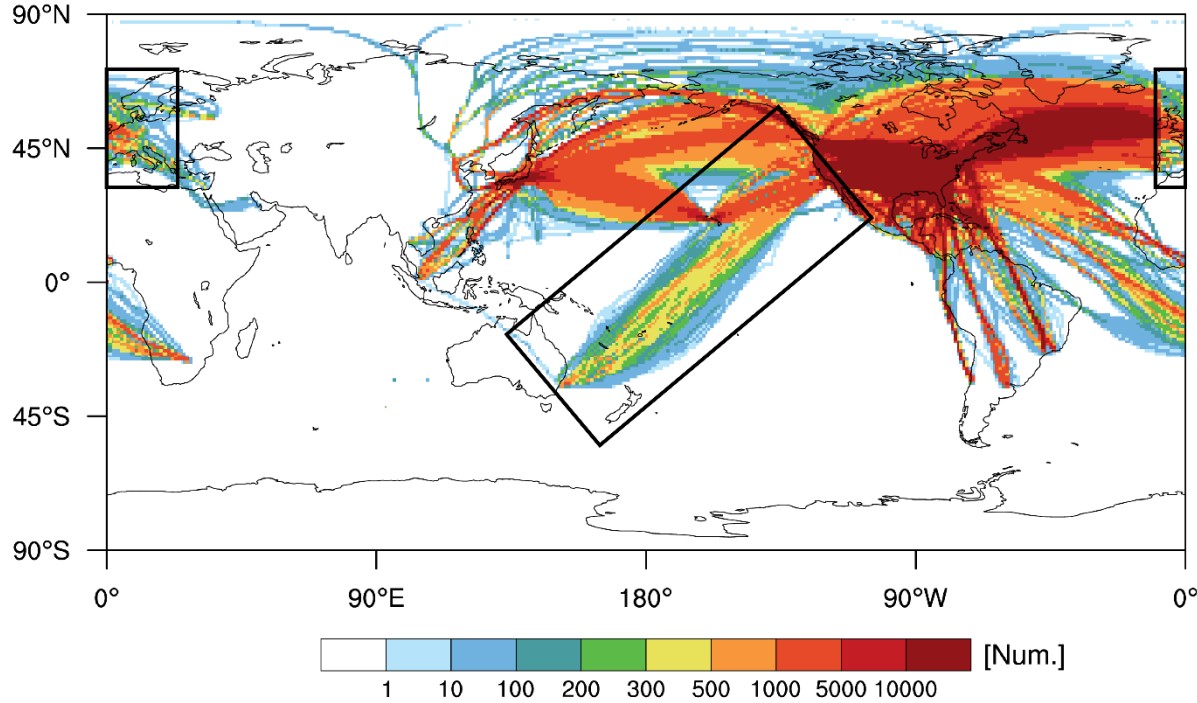

**Figure 10.** Horizontal distribution of the number of USEDR data at altitudes above 15 kft, accumulated within a 1°×1° horizontal grid box from October 2015 to September 2018. The regions of Europe and the trans-Pacific Ocean are indicated by rectangles.

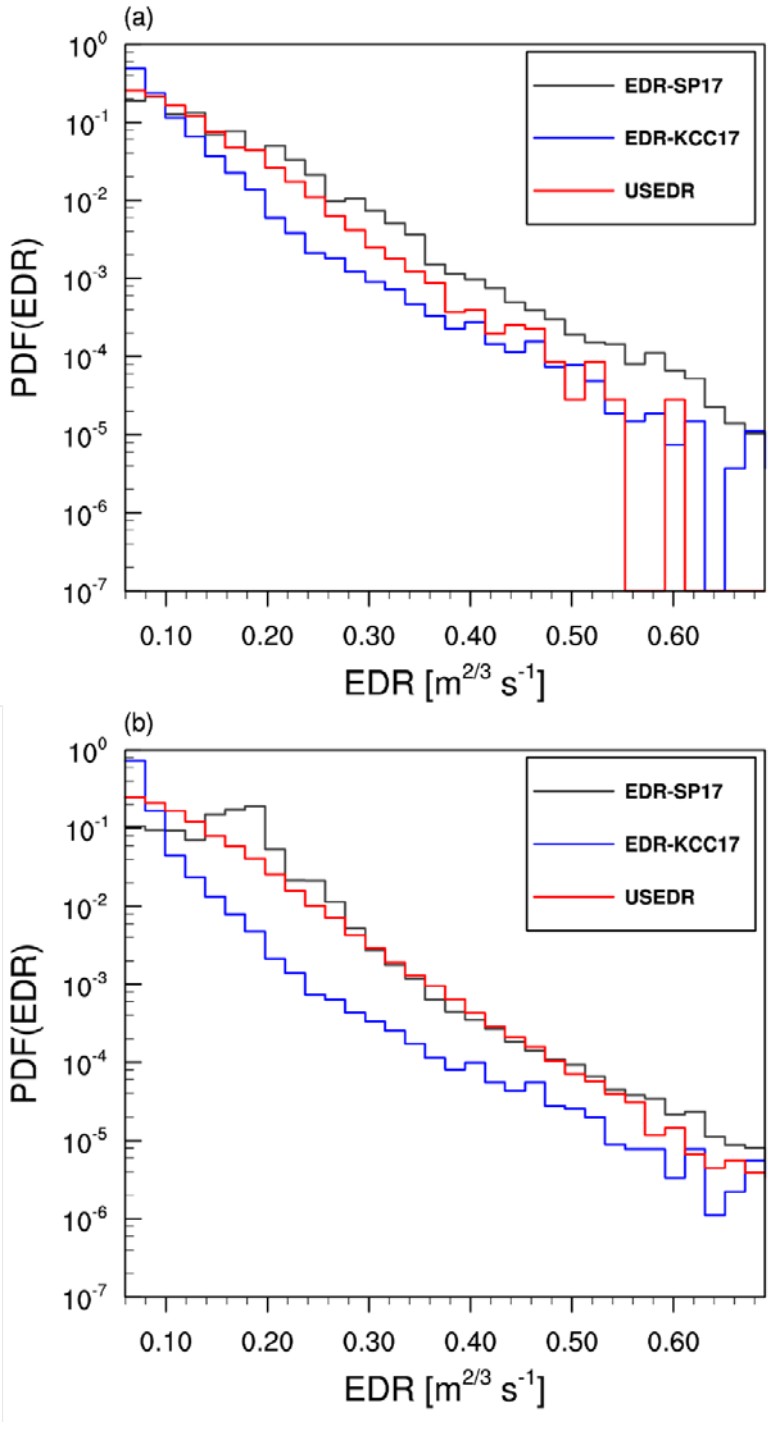

**Figure 11.** The PDFs of EDR-SP17 (black line), EDR-KCC17 (blue line), and USEDR (red line) at altitudes above 15 kft from October 2015 to September 2018 over (a) Europe and (b) the trans-Pacific Ocean routes indicated in Fig. 10.