# Peer review of "Retrieval of Eddy Dissipation Rate from Derived Equivalent Vertical Gust included in Aircraft Meteorological Data Relay (AMDAR)"

_Atmospheric Measurement Techniques, 2019_

## Referee Comment (RC1) · Anonymous Referee #1 · 16 Dec 2019

This study attempts to map the values of a turbulence diagnostic (derived equivalent vertical gust–DEVG) to a standard measure of turbulence intensity (cube root of the eddy dissipation rate–EDR). The motivation for obtaining such a mapping is that a limited fraction of commercial aircraft report either EDR or DEVG; therefore, establishing a correspondence between the two would allow turbulence data to be collected, studied, and exploited over an expanded global arena using a single, consistent metric–the aircraft-type-independent EDR.

The derived relationships are based on statistical comparisons of mutually exclusive data sets for DEVG and EDR over specific world regions. Although reasonable rela-

tionships result from the analysis, because DEVG and EDR measurements were not made simultaneously on the same flights, care must be taken to assure that apples are being compared to apples. Specific comments on this and other issues are given below.

Major Comments

1. Section 2.2: QC procedures. Quality control is absolutely critical for the DEVG data. Fits to the PDFs of DEVG (e.g., Figure 9) is at the core of this analysis. The PDF tails are very sensitive to small changes in the bin counts. However, these tails are where the raw DEVG have quite large bin counts in some regions that manifest as secondary modes in the PDF. The post-QC PDF tail is the residual of the difference between fairly large numbers. Thus, it is important to justify the QC procedure.

a. Why are there so many invalid DEVG values? Are there documented case studies that show how these errors occur? And why do they occur primarily in certain regions?

b. Related to (a), can you provide physical justifications for the four steps of the QC procedure?

c. Page 5, lines 29-30 state that the threshold values used in the QC steps are empirically determined. This empirical process needs to be explained clearly in detail. How can you tell that too few or too many reports were not removed? This is crucial, because errors in this process directly affect the tails of the DEVG PDFs.

2. Is parsing DEVG PDFs by northern or southern hemisphere the most meaningful and useful classification? There are reasons why the PDFs might differ for flights (a) over land vs. over ocean, (b) at different altitudes, (c) during different seasons, (d) during day vs. night, (e) in different |latitude| bands, etc.

3. As noted in p. 2, line 22, DEVG estimates may be inaccurate during ascent or descent, and, thus, the data at cruise altitudes (> 15 kft) only are used. However, even above 15 kft, aircraft can change altitudes and direction that could affect the

measurements. Why not restrict the use of data by only accepting estimates made during straight-and-level flight?

4. Figure 9 (and explanation in p. 7, lines 26-29). How do you justify throwing out some of the points in the PDFs for the fitting procedure?

5. In Figures 1 and 8, there is an oddly abrupt change in the data count right around the equator over the central Pacific. There is much more data across a wider swath south of the equator. Is this real? What is the cause of this sharp transition?

Minor Comments

1. DEVG and EDR have different units. Is there a physical basis on which to make a unit conversion? Or is there an explanation of why it is acceptable to ignore the difference in units?

2. Page 6, lines 6-7: It's not very informative that some MOD and SEV turbulence reports coincide with a high Ellrod1 index. Unless a statistical analysis is conducted to show a meaningful correlation, this remark should be omitted.

---

## Referee Comment (RC2) · Anonymous Referee #3 · 10 Jan 2020

General comments:

This paper investigates two methods of converting derived equivalent vertical gust (DEVG) turbulence measurements to the preferred ICAO standard turbulence metric eddy dissipation rate (EDR), using 3 years of archived AMDAR measurements. The two methods explored were proposed in two previous studies. The original DEVG measurements were subjected to a comprehensive quality control process which is described in detail. The accuracy of the resulting converted EDR values were examined by comparing them statistically to in-situ EDR turbulence measurements over two regions: over Europe and over the trans-Pacific Ocean area. The whole process is well described and discussed. The results of this study would enable

a wider range of homogenized aircraft observations of turbulence to be available for development and research work. This would aid the development of turbulence forecasts and enable the construction of an upper-level turbulence climatology over a much larger area of the globe.

In general, the paper is well written and well organised, and the results are of considerable interest. I therefore recommend that this manuscript should be accepted for publication with (very) minor revisions.

Specific comments:

Pg 8 Line 8: I didn't follow why only one of the equations in the best-fit function was used, rather than the correct equation for the aircraft type recording the DEVG. Is it that the aircraft type was missing from some of the observations in the dataset, so you needed to choose one equation?

Technical corrections:

Pg 1, line 12: . . .in the AMDAR **data** archived. . .

Pg1, line 15: The first method **remaps** the DEVG. . .

Pg 1, line 16: . . .while the second one **uses** the best-fit curve. . .

Pg1, line 16: "developed in the previous study". Which previous study was this (I don't think it's been mentioned yet)? Perhaps this part should be deleted, or written as: . . .developed in **a** previous study."

Pg 3, line 1: some aircraft of **a** Hong-Kong based airline.

Pg 3, line 10: Because **the** two aforementioned turbulence metrics. . .

Actually, I'd re-write this sentence as "As these two turbulence metrics. . ." which sounds clearer?

Pg3, line 10: different **airlines**

Pg 3, line 14: This may be better worded as "This will lead to improvements in the verification of. . . " ?

Pg 3, line 14: "as well as global climatology of aviation turbulence". This would be better as "as well as aid the construction of a global climatology of aviation turbulence", or similar?

Pg 3, line 19: either "some aircraft of **a** Hong-Kong based airline" or "some aircraft of **the** Hong-Kong based airline"

Pg 3, line 20: (39 months from February 2011 to April 2014) **of** data.

Pg 3 line 25: delete the hyphen between "(NOAA)" and "archives"

Pg 4, lines 9 -11: I would switch the first two sentences around, so it is something like: "The data before the QC procedures have been applied are referred to as the raw DEVG in the current study. Figure 1 shows the horizontal distribution of the number of raw DEVG data collected over 36 months (from October 2015 to September 2018) above 15kft accumulated within a 1°x1° horizontal box. The raw DEVG covers a large portion of the SH...."

Pg 4, line 13: "this raw DEVG can complement the SH turbulence information" I didn't follow this... the raw DEVG data complements the in-situ data (which mainly covers the NH), as it provides coverage over the SH. I think this line just needs re-wording?

Pg 5, line 11: I would replace "That is, because" with "Since" ?

Pg5, line 23: I would replace "Applying the aforementioned QC procedures" with "Applying these QC procedures"

Pg 7, lines 18 to 20: I got a bit confused with the first half of this sentence. Do you mean that there's a choice of several values of C1 and C2 for the altitude ranges in this study? And if so, isn't there a choice of 4 values? Or did I misunderstand something? I would also re-word the part in brackets so it is shorter and simpler – e.g. ... for three altitude ranges (> 0ft, 20-45ft, 10-20ft and 20-45ft)

Pg 10, line 10: "homogenized global turbulence **dataset**" or "homogenized global turbulence **archive**"

References section:
There were several references listed here which I couldn't find in the contents of the

paper:
Gultepe et al. (2019)
Tvaryanas (2003)
Warner (2013)
Williams (2017)

---

## Author Comment (AC2) · 4 Feb 2020

**[ Responses to the Comment by the Anonymous Referee #3 ]**

>> We deeply appreciate the referee#3 for providing constructive comments. The manuscript is revised following the comments below.

**General comments:**

This paper investigates two methods of converting derived equivalent vertical gust (DEVG) turbulence measurements to the preferred ICAO standard turbulence metric eddy dissipation rate (EDR), using 3 years of archived AMDAR measurements. The two methods explored were proposed in two previous studies. The original DEVG measurements were subjected to a comprehensive quality control process which is described in detail. The accuracy of the resulting converted EDR values were examined by comparing them statistically to in-situ EDR turbulence measurements over two regions: over Europe and over the trans-Pacific Ocean area. The whole process is well described and discussed. The results of this study would enable a wider range of homogenized aircraft observations of turbulence to be available for development and research work. This would aid the development of turbulence forecasts and enable the construction of an upper-level turbulence climatology over a much larger area of the globe. In general, the paper is well written and well organised, and the results are of considerable interest. I therefore recommend that this manuscript should be accepted for publication with (very) minor revisions.

**Specific comments:**

1) Page 8, Line 8: I didn't follow why only one of the equations in the best-fit function was used, rather than the correct equation for the aircraft type recoding the DEVG. Is it that the aircraft type was missing from some of the observations in the dataset, so you needed to choose one equation?

The aircraft-related information, such as aircraft type and tail number, is limited in the Aircraft Meteorological Data Relay (AMDAR) dataset. Kim et al. (2017) showed that the two turbulence indicators [the cube root of eddy dissipation rate (EDR) and derived equivalent vertical gust (DEVG)] from Boeing aircraft have higher correlation than those from Airbus aircraft, which is related to differences in the number of decimals and sampling frequency of

the recorded variables for each aircraft type [Table 1 of Kim et al. (2017)]. In this regard, we decided to use the best-fit curve from the Boeing aircraft. This statement is included in the original manuscript.

**Technical corrections:**

1) Page 1, Line 12: … in the AMDAR **data** archived …
The sentence is modified as suggested. [Page 1, Line 12]

2) Page 1, Line 15: The first method **remaps** the DEVG …
The sentence is modified as suggested. [Page 1, Line 15]

3) Page 1, Line 16: … while the second one **uses** the best-fit curve …
The sentence is modified as suggested. [Page 1, Line 16]

4) Page 1, Line 16: "developed in the previous study". Which previous study was this (I don't think it's been mentioned yet)? Perhaps this part should be deleted, or written as: … developed in **a** previous study."
The sentence is modified as suggested. [Page 1, Line 16]

5) Page 3, Line 1: some aircraft of **a** Hong-Kong based airline.
The sentence is modified as suggested. [Page 3, Line 1]

6) Page 3, Line 10: "Because **the** two aforementioned turbulence metrics …". Actually, I'd re-write this sentence as "As these two turbulence metrics …" which sounds clearer?
The sentence is modified as suggested. [Page 3, Line 10]

7) Page 3, Line 10: different **airlines**
The sentence is modified as suggested. [Page 3, Line 10]

8) Page 3, Line 14: This may be better worded as "This will lead to improvements in the verification of …"?

The sentence is modified as suggested. [Page 3, Line 13]

9) Page 3, Line 14: "as well as global climatology of aviation turbulence". This would be better as "as well as aid the construction of a global climatology of aviation turbulence", or similar ?

The sentence is modified as suggested. [Page 3, Line 14]

10) Page 3, Line 19: either "some aircraft of **a** Hong-Kong based airline" or "some aircraft of **the** Hong-Kong based airline"

The sentence is modified as suggested. [Page 3, Line 19]

11) Page 3, Line 20: (39 months from February 2011 to April 2014) **of** data.

The sentence is modified as suggested. [Page 3, Line 20]

12) Page 3, Line 25: delete the hyphen between "(NOAA)" and "archives".

The hyphen between "NOAA" and "archives" is deleted in the revised manuscript.

13) Page 4, Lines 9-11: I would switch the first two sentences around, so it is something like: "The data before the QC procedures have been applied are referred to as the raw DEVG in the current study. Figure 1 shows the horizontal distribution of the number of raw DEVG data collected over 36 months (from October 2015 to September 2018) above 15 kft accumulated within a 1°×1° horizontal box. The raw DEVG covers a large portion of the SH …".

The sentence is modified as suggested. [Page 4, Line 8-10]

14) Page 4, Line 13: "this raw DEVG can complement the SH turbulence information" I didn't follow this … the raw DEVG data complements the in-situ data (which mainly covers the NH), as it provides coverage over the SH. I think this line just needs re-wording?

We would like to emphasize that the raw DEVG data will complement the Southern Hemisphere (SH) turbulence information, given that the in situ EDR data covered most the Northern Hemisphere (NH) (Figure 10 of the original manuscript) and did not provide coverage over most SH regions. The sentence is modified as suggested. [Page 4, Line 12-13]

15) Page 5, Line 11: I would replace "That is, because" with "Since"?

The sentence is modified as suggested. [Page 5, Line 15]

16) Page 5, Line 23: I would replace "Applying the aforementioned QC procedures" with "Applying these QC procedures"

The sentence is modified as suggested. [Page 5, Line 28]

17) Page 7, Lines 18-20: I got a bit confused with the first half of this sentence. Do you mean that there's a choice of several values of $C_1$ and $C_2$ for the altitude ranges in this study? And if so, isn't there a choice of 4 values? Or did I misunderstand something? I would also re-word the part in brackets so it is shorter and simpler – e.g., … for three altitude ranges (> 0 ft, 20-45 kft, 10 – 20 kft and 20 – 45 kft).

To avoid any confusion, the sentence is modified. [Page 8, Line 1-2]

18) Page 10, Line 19: "homogenized global turbulence **dataset**" or "homogenized global turbulence **archive**"

The sentence is modified as suggested. [Page 11, Line 4]

**References section:**

1) There were several references listed here which I couldn't find in the contents of the paper: Gultepe et al. (2019), Tvaryanas (2003), Warner (2013), Williams (2017).

The aforementioned references are deleted in the revised manuscript.

**References**

Kim, S.-H., H.-Y. Chun, and P. W. Chan, 2017: Comparison of turbulence indicators obtained from in situ flight data. *J. Appl. Meteor. Climatol.*, **56,** 1609-1623, doi:10.1174/JAMC-D-16-0291.1

Sharman, R. D., and J. Pearson, 2017: Prediction of energy dissipation rates for aviation turbulence. Part I: Forecasting nonconvective turbulence. *J. Appl. Meteor. Climatol.*, **56,** 317-337, doi:10.1175/JAMC-D-16-0205.1.

---

## Author Response (AR1)

**Response to Referees' Comments**

**S.-H. Kim, H.-Y. Chun, J.-H. Kim, R. D. Sharman, and M. Strahan**

**February 8, 2020**

Dear editor and reviewers,

We received reviews for our manuscript "Retrieval of eddy dissipation rate from derived equivalent vertical gust included in aircraft meteorological data relay (AMDAR)". All reviews have been beneficial and made us aware of some major and minor points which had to be reflected. We, the authors, are therefore thankful for their contribution to further improve the manuscript's quality. We carefully addressed all comments and tried our best to improve the manuscript based on the suggestions and comments. We answer questions in the following paragraphs. Below, we indicate the original comment of the respective reviewer in blue and our answer is denoted in black. In addition, we provide a tracked-changes version of the manuscript.

Sincerely,

Hye-Yeong Chun

**[ Responses to the Comment by the Anonymous Referee #1 ]**

>> We deeply appreciate the referee#1 for providing constructive comments. The manuscript is revised following the comments below.

This study attempts to map the values of a turbulence diagnostic (derived equivalent vertical gust-DEVG) to a standard measure of turbulence intensity (cube root of the eddy dissipation rate-EDR). The motivation for obtaining such a mapping is that a limited fraction of commercial aircraft report either EDR or DEVG; therefore, establishing a correspondence between the two would allow turbulence data to be collected, studied, and exploited over an expanded global area using a single, consistent metric-the aircraft-type-independent EDR. The derived relationships are based on statistical comparisons of mutually exclusive data sets for DEVG and EDR over specific world regions. Although reasonable relationships result from the analysis, because DEVG and EDR measurements were not made simultaneously on the same flights, care must be taken to assure that apples are being compared to apples. Specific comments on this and other issues are given below.

**Major Comments:**

1) Section 2.2: QC procedures. Quality control is absolutely critical for the DEVG data. Fits to the PDFs of DEVG (e.g., Figure 9) is at the core of this analysis. The PDF tails are very sensitive to small changes in the bin counts. However, these tails are where the raw DEVG have quite large bin counts in some regions that manifest as secondary modes in the PDF. The post-QC PDF tail is the residual of the difference between fairly large numbers. Thus, it is important to justify the QC procedure.

a. Why are there so many invalid DEVG values? Are there documented case studies that show how these errors occur? And why do they occur primarily in certain regions?

As far as we know, there is no official document on the quality issue of the derived equivalent vertical gust (DEVG) data. Instead, in the Aircraft Meteorological Data Relay (AMDAR) observing system newsletters published by the World Meteorological Organization (WMO), the horizontal distribution of the DEVG has been reported since April 2016

(https://sites.google.com/a/wmo.int/amdar-news-and-events/newsletters/volume-11-april-2016). As in Fig. A1, the DEVG indicates relatively large values over some regions of Australia, New Zealand, and Europe, which is a consistent distribution with the current study. It is noted that the WMO newsletter did not mention the quality of the DEVG. The possible reasons of suspicious DEVG values can be a power loss of electrical power contactor and a bug in the DEVG initialization logic, which is related to intermittently added 1-$g$-bias, where $g$ is the acceleration due to gravity (D. Body, personal communication, November, 2019). Considering that the DEVG data merged point samples from many kinds of flight over the globe, and time series of the DEVG and other recorded variables are not available, it is difficult to clearly identify the reason of suspicious DEVG values. A further investigation of the quality control (QC) procedures of the DEVG remains for future work, after obtaining the time series of recorded variables. A statement is included in the revised manuscript. [Page 6, Line 6-8]

[Figure]

Figure A1. The horizontal distribution of the DEVG values for one day on (a) 21 September 2018 and (b) 31 March 2019, adapted from the WMO AMDAR observing system newsletter (https://sites.google.com/a/wmo.int/amdar-news-and-events/newsletters/volume-16-october-2018 and https://sites.google.com/a/wmo.int/amdar-news-and-events/newsletters/volume-17-april-2019).

b. Related to (a), can you provide physical justifications for the four steps of the QC procedure?
As done in previous studies (e.g., Gill 2014; Meneguz et al. 2016; Kim et al. 2017), the minimum flight time is considered, which is set to approximately one hour within an individual file that has the same flight tail number. As the time series of recorded variables are not available, we adopt an approach that uses a cluster of the DEVG data within a certain spatiotemporal window to increase a confidence of a turbulence event. An early version of the QC procedures in the current study is designed based on those by Gill (2014) and Meneguz et

al. (2016) that used the Global Aircraft Data Set. These QC procedures to the Australian AMDAR data for four years (from 2011 to 2014) are revised through interactive discussions with two scientists affiliated in the Met Office (D. Turp and P. G. Gill, personal communications, February 2015). Again, these QC procedures are revised based on active discussions with scientists and forecasters associated in the Aviation Weather Center/NCEP (personal communications, from June to August 2018). During the QC procedures, we checked horizontal distributions of the raw and QC'd DEVG data when all moderate (MOD) and severe (SEV) turbulence events are reported. At least in the current study, the irrelevant turbulence events are discarded and a probability density function (PDF) of the DEVG follows the lognormal distribution, which is consistent with previous studies (e.g., Nastrom and Gage 1985; Frehlich 1992; Frehlich and Sharman 2004; Sharman et al. 2014; Kim et al. 2017). A statement is included in the revised manuscript. [Page 5, Line 29-33; Page 6, Line 17-19]

c. Page 5, lines 29-30 state that the threshold values used in the QC steps are empirically determined. This empirical process needs to be explained clearly in detail. How can you tell that too few or too many reports were not removed? This is crucial, because errors in this process directly affect the tails of the DEVG PDFs.

As mentioned above, the validity of QC procedures was discussed with the experts of the aircraft-based observations. In the raw DEVG data, only SEV (MOD) turbulence events were reported without MOD (LGT) turbulence event (Fig. A2), which is the most suspicious distribution. We checked all SEV and MOD turbulence events and excluded irrelevant turbulence events by comparing surrounding turbulence events. It is confirmed that the PDF of QC'd DEVG follows a lognormal distribution, as shown in Kim et al. (2017). A further investigation of the QC procedures of the DEVG remains to be accomplished in the future, after obtaining the time series of recorded variables. A statement is included in the revised manuscript. [Page 5, Line 29-33; Page 6, Line 17-19]

[Figure]

Figure A2. The horizontal distribution of the raw DEVG data for one hour of (a) 1000-1059 UTC 5 October 2015 and (b) 0100-0159 UTC 11 October 2016. The null (DEVG < 2 m s$^{-1}$), MOD ($4.5 \leq$ DEVG < 9 m s$^{-1}$), and SEV (DEVG $\geq$ 9 m s$^{-1}$) turbulence events are indicated as cyan, blue, and red circles, respectively.

2. Is parsing DEVG PDFs by northern or southern hemisphere the most meaningful and useful classification? There are reasons why the PDFs might differ for flights (a) over land vs. over ocean, (b) at different altitudes, (c) during different seasons, (d) during day vs. night, (e) in different latitude bands, etc.

As discussed in comment #5, the reporting frequency is changed when the aircraft passes the equator, which is indicated as an abrupt change in a data count around the equator. Considering the geographical difference in the data count, the PDFs over the Northern Hemisphere (NH) and Southern Hemisphere (SH) were computed and examined in the current study. During the revision process, we examined the PDF of the DEVG over land and ocean, different altitude ranges (15-25 kft, 25-35 kft, and 35-45 kft), seasons (spring, summer, autumn, and winter), day and night, and different latitude bands of a spacing of 20°. The mean and standard deviation of the natural logarithm of the DEVG (Table A1) are not significantly changed for aforementioned conditions, except that those in latitudes equatorward of 30° are clearly smaller than those poleward of 30°. This statement is included in the revised manuscript. [Page 8, Line 12-16]

Table A1. Values of the mean and standard deviation (SD) of the natural logarithms of the DEVG. The unit is m s$^{-1}$.

(a) Land or Ocean

|  | Land | Ocean |
| --- | --- | --- |
| Mean | -1.1610 | -1.1290 |
| SD | 0.7181 | 0.7881 |

(b) Altitude bands

|  | 15-25 kft | 25-35 kft | 35-45 kft |
| --- | --- | --- | --- |
| Mean | -1.8025 | -1.3114 | -0.8702 |
| SD | 1.1149 | 0.7974 | 0.6106 |

(c) Seasons

| | Spring | Summer | Autumn | Winter |
|---|---|---|---|---|
| Mean | -1.1985 | -1.2953 | -1.2812 | -1.2448 |
| SD | 0.7776 | 0.8278 | 0.8124 | 0.8005 |

(d) Daytime or Nighttime

| | Daytime | Nighttime |
|---|---|---|
| Mean | -1.2298 | -1.2185 |
| SD | 0.7488 | 0.8277 |

(e) Latitude bands

| | 70°S-50°S | 50°S-30°S | 30°S-10°S | 10°S-10°N | 10°N-30°N | 30°N-50°N | 50°N-70°N |
|---|---|---|---|---|---|---|---|
| Mean | -1.9345 | -1.2325 | -0.6136 | -0.8136 | -0.4624 | -1.6851 | -1.8362 |
| SD | 0.9532 | 0.5945 | 0.4269 | 0.7049 | 0.5833 | 1.3877 | 1.2280 |

3. As noted in p. 2, line 22, DEVG estimates may be inaccurate during ascent or descent, and, thus, the data at cruise altitudes (> 15 kft) only are used. However, even above 15 kft, aircraft can change altitudes and direction that could affect the measurements. Why not restrict the use of data by only accepting estimates made during straight-and-level flight?

Thank you for pointing out this rather important issue. Unfortunately, the AMDAR data used in the current study do not provide the phase of flight which indicates 'level flight', 'ascending', and 'descending'. As in this study, Kim and Chun (2016) used 15 kft as the lower limit of altitude based on the time series of the DEVG and several variables recorded in the in situ flight data recorders. This statement is included in the revised manuscript. [Page 4, Line 19-21]

4. Figure 9 (and explanation in p. 7, lines 26-29). How do you justify throwing out some of the points in the PDFs for the fitting procedure?

At the highest bins, there are not enough data for reliable lognormal fits, while at the lowest bins, instrument noise may be affecting the result and the small DEVG values corresponding to nonturbulent conditions are not of practical interest. This statement is included in the revised manuscript. [Page 8, Line 8-10]

5. In Figures 1 and 8, there is an oddly abrupt change in the data count right around the equator over the central Pacific. There is much more data across a wider swath south of the equator. Is this real? What is the cause of this sharp transition?

The abrupt change in the data count around the equator can be found in the WMO AMDAR observing newsletter (Figs. A1 and A3), which is consistent with the current study (Fig. 1 of the original manuscript). For some reasons, when the aircraft passes the equator, the reporting frequency is changed from relatively lower to higher or higher to lower (Figs. A3 and A4). This difference in the reporting frequency between the NH and SH brings a sharp transition between two hemispheres. The abrupt transition across the equator may be related to systematic settings in aircraft-to-ground reporting during navigation. As the raw DEVG data indicate the abovementioned features, we would like to mention that this can be one of the characteristics of the DEVG data included in the AMDAR data and therefore the DEVG data can provide much more turbulence observations over the SH. This statement is included in the revised manuscript. [Page 4, Line 16-19]

[Figure]

Figure A3. The horizontal distribution of the number of the DEVG data, accumulated within a 50×50 km grid box from 1 September to 1 October 2019. Adapted from the WMO AMDAR observing system newsletter (https://sites.google.com/a/wmo.int/amdar-news-and-events/newsletters/volume-18-october-2019).

[Figure]

Figure A4. The horizontal location of the DEVG reports of the same flight tail number for one day (5 October 2015).

1. DEVG and EDR have different units. Is there a physical basis on which to make a unit conversion? Or is there an explanation of why it is acceptable to ignore the difference in units? As written in the original manuscript (section 3.1), a simple mapping equation from a certain turbulence diagnostics $D$ to the EDR was proposed by Sharman and Pearson (2017) using the observed in situ EDR data. First, turbulence diagnostics $D$ and the EDR are standardized as:

$$X_1 = \frac{\ln(D^*) - \langle \ln(EDR) \rangle}{SD\ln(EDR)},$$  (1)

$$X_2 = \frac{\ln(D) - \langle \ln(D) \rangle}{SD\ln(D)},$$  (2)

where the angle brackets indicate an ensemble mean, SD is a standard deviation, $D^*$ is the remapped EDR value corresponding to the raw turbulence diagnostics, and $EDR$ is the in situ EDR values. Second, two standardized values, $X_1$ and $X_2$, are set to be equal and Eqs. (1) and (2) can be combined as:

$$\ln(D^*) = \langle \ln(EDR) \rangle + \frac{SD\ln(EDR)}{SD\ln(D)} \{ \ln(D) - \langle \ln(D) \rangle \}.$$  (3)

Therefore, Eq. (3) is written as the simple form as:

$$\ln(D^*) = \ln(EDR) = a + b\ln(D),$$
$$a = \langle \ln(EDR) \rangle - b\langle \ln(D) \rangle, \text{ and}$$  (4)
$$b = \frac{SD\ln(EDR)}{SD\ln(D)}.$$

In the current study, the turbulence diagnostics $D$ is replaced with the DEVG value as:

$$\ln(DEVG^*) = \ln(EDR) = a + b\ln(DEVG),$$
$$a = \langle \ln(EDR) \rangle - b\langle \ln(DEVG) \rangle, \text{ and}$$  (5)
$$b = \frac{SD\ln(EDR)}{SD\ln(DEVG)}.$$

where $DEVG^*$ is the remapped EDR value corresponding to the DEVG value included in the AMDAR data. In this regard, it is acceptable to ignore a difference in units, although the EDR is a direct turbulence intensity metric, while the DEVG is not a direct turbulence intensity metric but a gust-load transfer factor.

2. Page 6, lines 6-7: It's not very informative that some MOD and SEV turbulence reports coincide with a high Ellrod1 index. Unless a statistical analysis is conducted to show a meaningful correlation, this remark should be omitted.

We agree with the reviewer and the sentence is deleted in the revised manuscript.

The aforementioned references are deleted in the revised manuscript.

**References**

[revised manuscript text omitted]

---

## Author Response (AR2)

**[ Responses to the Comment by the Anonymous Referee ]**

>> We deeply appreciate the referee for providing constructive comments. The manuscript is revised following the comments below.

I think the paper is in quite good shape after the changes made in response to the initial round of comments. Now I have just a couple of recommendations, and a list of incidental language usage corrections.

1. In the response to Referee #1's major comment 1a, there is some information that I did not see in the revised paper. Specifically, the statement: "[P]ossible reasons [for] suspicious DEVG values [could] be power loss [at the] electrical contact[s] and a bug in the DEVG initialization logic, which is related to [an] intermittently added 1-g-bias, where g is the acceleration due to gravity (D. Body, personal communication)". This information would be useful to include in the paper.

A statement is included as suggested. [Page 6, Line 8-10]

2. Page 2, Line 20: Rather than saying that the DEVG uncertainties are assumed to be negligible, it would be more informative to quote the actual figure from the AMDAR manual (3 to 4% typically, 10 to 12 % in the extreme).

The statement is included as suggested. [Page 2, Line 20]

**Language corrections:**

1. Page 1, Line 20: "statistics by" → "statistics obtained by".

The sentence is modified as suggested. [Page 1, Line 20]

2. Page 1, Line 28: "routine, therefore PIREPs" → "routine; therefore, PIREPs".

The sentence is modified as suggested. [Page 1, Line 28]

3. Page 2, Line 6: "metrics are" → "metrics were".

The sentence is modified as suggested. [Page 2, Line 6]

4. Page 3, Line 2: "adopted the" → "adopted a"; "and different" → "and a different".

The sentence is modified as suggested. [Page 3, Line 2]

5. Page 3, Line 3: "between two" → "between the two".

The sentence is modified as suggested. [Page 3, Line 3]

6. Page 3, Line 4: "metric detection" → "detection metric".

The sentence is modified as suggested. [Page 3, Line 4]

7. Page 4, Line 2: "NOAA include" → "NOAA that include".

The sentence is modified as suggested. [Page 4, Line 2]

8. Page 4, Line 13: "This reporting" → "The reporting".

The sentence is modified as suggested. [Page 4, Line 13]

9. Page 4, Line 15: "lower reporting time window" → "long reporting time windows".

The sentence is modified as suggested. [Page 4, Line 15]

10. Page 4, Line 16: "counts either" → "counts on either".

The sentence is modified as suggested. [Page 4, Line 16]

11. Page 5, Line 32: "to" → "for".

The sentence is modified as suggested. [Page 5, Line 32]

12. Page 6, Line 8: "reason of" → "reason for".

The sentence is modified as suggested. [Page 6, Line 8]

13. Page 7, Line 5: Delete "forecast".

The word "forecast" is deleted as suggested.

14. Page 7, Line 11: "the lognormal property of" → "a lognormal property for".

The sentence is modified as suggested. [Page 7, Line 12]

15. Page 8, Line 5: Delete "for".

The preposition "for" is deleted as suggested.

16. Page 8, Line 14: "bands of" → "bands with".

The sentence is modified as suggested. [Page 8, Line 16]

17. Page 8, Line 15: "for" → "for the".

The sentence is modified as suggested. [Page 8, Line 17]

18. Page 8, Line 22: "which developed the best-fit" → "based on the best-fit".

The sentence is modified as suggested. [Page 8, Line 24]

19. Page 8, Line 29: Delete "for".

The preposition "for" is deleted as suggested.

20. Page 9, Line 16: "with" → "as".

The sentence is modified as suggested. [Page 9, Line 16]

21. Page 10, Line 23: "the lognormal property of" → "a lognormal property for".

The sentence is modified as suggested. [Page 10, Line 23]

22. Page 10, Line 23: "proposed the" → "proposed a".

The sentence is modified as suggested. [Page 10, Line 23]

23. Page 10, Line 24: "on the" → "on a".

The sentence is modified as suggested. [Page 10, Line 24]

24. Page 10, Line 28: "The robust" → "A robust".

The sentence is modified as suggested. [Page 10, Line 28]

25. Page 11, Line 5: "of global" → "of a global".

The sentence is modified as suggested. [Page 11, Line 5]

[revised manuscript text omitted]